# Universal machine learning aided synthesis approach of two-dimensional perovskites in a typical laboratory

Yilei Wu[1,4], Chang-Feng Wang[2,4], Ming-Gang Ju [1,4] ✉, Qiangqiang Jia[2], Qionghua Zhou[1], Shuaihua Lu[1], Xinying Gao[1], Yi Zhang [2] ✉ & Jinlan Wang [1,3] ✉

The past decade has witnessed the significant efforts in novel material discovery in the use of data-driven techniques, in particular, machine learning (ML). However, since it needs to consider the precursors, experimental conditions, and availability of reactants, material synthesis is generally much more complex than property and structure prediction, and very few computational predictions are experimentally realized. To solve these challenges, a universal framework that integrates high-throughput experiments, a priori knowledge of chemistry, and ML techniques such as subgroup discovery and support vector machine is proposed to guide the experimental synthesis of materials, which is capable of disclosing structure-property relationship hidden in high-throughput experiments and rapidly screening out materials with high synthesis feasibility from vast chemical space. Through application of our approach to challenging and consequential synthesis problem of 2D silver/bismuth organic-inorganic hybrid perovskites, we have increased the success rate of the synthesis feasibility by a factor of four relative to traditional approaches. This study provides a practical route for solving multidimensional chemical acceleration problems with small dataset from typical laboratory with limited experimental resources available.

The discovery of advanced functional materials has the power to help combat the major global challenges facing humanity[1,2]. However, materials synthesis is a typical complex, multidimensional challenge that requires experts to evaluate various reaction conditions, such as precursors, additives, solvents, concentration, and temperature[3]. Owing to an inherent limitation based on the availability and provision of chemical precursors and experimental instruments, synthetic chemists can only evaluate a small subset of these conditions during a standard optimization campaign in a typical and simple laboratory. Likewise, the exploration of conditions is often left in the hands of predefined optimal design, limited literature on solid-state synthetic reactions, and the experience of chemists. The fundamental challenges

associated acceleration of material synthesis in a typical laboratory with limited experimental support is an urgent concern[4].

Data-driven machine learning (ML) techniques have emerged as a powerful tool for the design and discovery of advanced materials in the past few years[5–8]. These techniques can excavate the structure–property relationship and uncover in-depth physical insights from existing data, and then make rapid predictions for properties of unexplored materials[9,10]. Although ML techniques have been successfully utilized in data-rich systems such as predicting the formability and properties of materials[11–13], the utilization of these techniques to guide the experimental synthesis of new materials has still been limited[14–16]. The major challenge is the acquisition of big and

[1]Key Laboratory of Quantum Materials and Devices of Ministry of Education, School of Physics, Southeast University, 211189 Nanjing, China. [2]Institute for Science and Applications of Molecular Ferroelectrics, Key Laboratory of the Ministry of Education for Advanced Catalysis Materials, Zhejiang Normal University, 321004 Jinhua, China. [3]Suzhou Laboratory, Suzhou, China. [4]These authors contributed equally: Yilei Wu, Chang-Feng Wang, Ming-Gang Ju. ✉e-mail: juming@seu.edu.cn; yizhang1980@seu.edu.cn; jlwang@seu.edu.cn

complete experimental synthesis data for conventional ML techniques. As an important source of material data, experimental synthesis data in literature exhibits a strong bias toward successful experiments, namely, materials that have been synthesized. The failed experiments are often recorded in the unpublic laboratory notebook, leading to the imbalanced distribution of experimental synthesis data. Another common source of material data, first-principles calculations, however, usually exhibit a large gap with actual experiments. Due to the discards of several factors impacting the synthesis stage, such as experimental conditions and availability of precursors, only a rather small fraction of theoretically designed materials have been synthesized experimentally. Very recently, a closed-loop automated synthesis framework based on ML techniques and robotic experimentation has proven to be efficient in accelerating the experimental synthesis process, coming with high experimental costs[17]. Moreover, many time-consuming experiments enable only the provision of small-scale datasets, which are incommensurate with conventional ML methods because of the inherent sparsity and imbalance of the available data[18]. Small datasets and imbalanced data distributions can easily bring about serious issues like overfitting, underfitting, and limited extrapolating abilities of ML models[19,20]. Several strategies have been proposed to address class imbalance problems based on over-sampling

and under-sampling method[21]. Although there are numerous attempts to address these challenges, a comprehensive ML framework suitable for unfaithful datasets in material science has not yet been established. Therefore, the development of a framework integrating ML techniques and small-scale experiments to rapidly accelerate the material synthesis process is especially important for branching out into new material space.

Two-dimensional hybrid organic–inorganic perovskites (2D HOIPs) have emerged as one of the most promising functional materials, with the benefits of enhanced environmental stability[22], superior optical properties[23–25], diverse electronic properties[26–28], and accessible and cost-effective fabrication[29,30]. Inspired by their excellent performance, there exists an ever-growing interest in developing novel, stable, and environmentally friendly 2D HOIP materials. To date, the design and discovery of new 2D perovskites heavily relies on the traditional trial-and-error method. With several millions of experimental available organic molecules and dozens of inorganic frameworks, the unexplored chemical space contains a large number of potential novel 2D HOIPs, making searches based on the traditional trial-and-error method frustratingly slow and expensive. One possible solution is to integrate small-scale perovskite synthesis experiments, non-learned representation approaches from knowledge of chemistry or mechanisms a priori[17], and innovative ML techniques. For instance, Sun et al. fabricated and characterized 73 unique perovskite-inspired compositions, and used ML techniques to classify compounds into 0D, 2D, and 3D structures[15]. Kirman et al. reported a high-throughput experimental framework with the aid of ML techniques for the discovery of new perovskite single crystals[14]. This strategy that combines small-scale high-throughput experiments with ML techniques points out a promising direction for new material discovery and improves the experimental efficiency in comparison with the trial-and-error method.

This work showcases the synthesis feasibility of 2D silver/bismuth (AgBi) iodide perovskites, which have been suggested for application on photodetectors[31], light-emitting diodes[32], and X-ray imagers[33]. We develop a framework combining small-scale high-throughput experiments, quantifying steric and topological properties of organic precursors, and ML techniques to rapidly screen 2D HOIPs with high synthesis feasibility (Fig. 1). The material dataset is acquired by performing high-throughput experiments, containing synthesis results of 80 tested amines, which can be divided into 14 succeeded and 66 failed synthesis experiments. In view of the interaction between inorganic layers and organic spacers of 2D perovskites, a set of informative features to quantify steric and topological properties of organic precursors is developed. With the aid of the subgroup discovery method, a region that is more favorable to form the 2D AgBi iodide perovskites is derived. Then an equation that can quantitatively evaluate the synthesis feasibility of 2D AgBi iodide perovskites is acquired by applying the ML techniques and 344 of 8406 organic spacers are predicted to hold the potential for the formation of 2D AgBi perovskites. Further interpretable ML technique, namely SHapley Additive exPlanations (SHAP) analysis, highlights the importance of molecular topology of organic spacers on the formation of 2D AgBi perovskites. In the end, 8 of 13 predicted 2D AgBi iodide perovskites with high synthesis feasibility are successfully synthesized, validating the good predictive ability of our ML-guided perovskite design strategy.

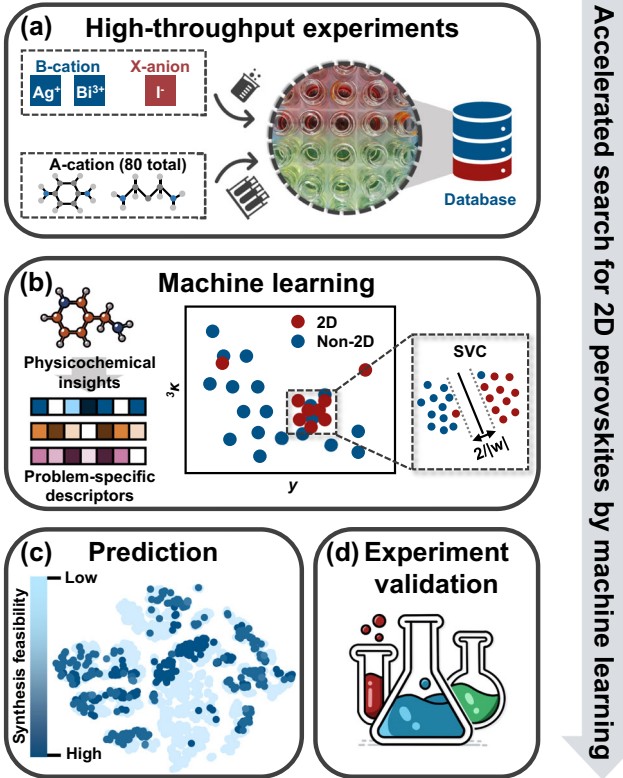

**Fig. 1 | Screening framework for two-dimensional silver/bismuth (2D AgBi) iodide perovskites.** The screening framework integrates high-throughput experiments, physicochemical insights, and ML techniques, each step of which is represented by a gray box. **a** Material database is acquired from high-throughput synthesis experiments, containing 14 positive samples and 66 negative samples. **b** Based on chemical intuition and machine learning (ML) techniques, a support vector classification (SVC) model to evaluate the synthesis feasibility of 2D AgBi iodide perovskites is developed. Here, $w$ represents the normal vector to the hyperplane. $^3\kappa$ and $y$ represent the third-order kappa shape index and width of molecules, respectively. **c** The synthesis feasibility of compounds in the prediction set is assessed and visualized by applying the t-distributed stochastic neighbor embedding (t-SNE) method[62]. **d** 13 predicted 2D perovskites with commercially available precursors are unbiased and selected to experimentally validate the reliability of our proposed equation.

## Results

### High-throughput synthesis experiments

The quality and quantity of training dataset is the cornerstone of the development of high-performance ML models. Regrettably, only a limited number of inorganic frameworks of 2D HOIPs have been experimentally realized. While the synthesis feasibility and properties of 2D HOIPs can be flexibly modulated through the use of various organic spacers during the material synthesis process, it is evident that the physicochemical properties of the organic spacers play a crucial

**Fig. 2 | Summary of high-throughput experimental synthesis results.** Organic spacers are classified into the box of "2D perovskite" and "non-2D perovskite" based on whether the 2D perovskite structure is formed.

role in determining the synthesis feasibility of 2D HOIPs. Previous studies[30] and our extensive laboratory experience have provided valuable chemical intuitions into the selection of organic spacers that are conducive to forming the 2D perovskite structure. To satisfy the charge neutrality condition, monovalent and divalent organic spacers are generally incorporated into 2D perovskites. Furthermore, these organic spacers should have moderate size to fit in the inorganic framework of 2D perovskites. Linear and cyclic organic spacers, whether aliphatic or aromatic, are found to be favorable for the formation of 2D perovskite structures. Taking into account organic spacers employed in previously reported 2D perovskites, along with the chemical intuitions mentioned above, and the commercial availability of amines, we have selected 79 promising amines for use in 2D AgBi iodide perovskite synthesis (Fig. 2).

To reduce experimental cost in this work, the same experimental conditions such as inorganic precursors, solvent, concentration, and temperature, are utilized in practice. High-throughput experimental results revealed that only 13 kinds of organic spacers can form 2D AgBi iodide perovskite structures, leading to the chemist intuition success rate of 16.4% (Supplementary Figs. 1 and 2, Supplementary Data 1). Based on the results of synthesis experiments, organic spacers are labeled as "2D perovskite" and "non-2D perovskite". The single-crystal structures of 13 synthesized 2D AgBi perovskites are obtained by single-crystal X-ray diffractometer, and the purity of bulk phases is confirmed by powder X-ray diffraction (PXRD) measurements (Supplementary Figs. 4 and 5). All synthesized 2D AgBi iodide perovskites show the typical single-layer structure, which can be further divided into Ruddlesden–Popper (RP) phase with the stoichiometry $A_4AgBiI_8$ (A = monovalent cation) or Dion–Jacobson (DJ) phase with the stoichiometry $A_2AgBiI_8$ (A = divalent cation) (Supplementary Tables 1–5). A-site organic cations are incorporated as spacers between inorganic layers, which are formed by alternating $AgI_6$ and $BiI_6$ octahedra. Metal cations (Ag and Bi) and iodine sit at the center and vertex of metal halide octahedra, respectively. Due to the avoidance of van der Waals interaction between organic spacer layers, 2D DJ perovskites with monolayer divalent A-site organic cations exhibit higher stability than 2D RP perovskites with bilayer monovalent A-site organic cations[22]. Moreover, the semiconducting properties of 13 synthesized 2D AgBi perovskites are further investigated by measuring ultraviolet–visible (UV–vis) diffuse reflectance spectroscopy. The gradually decreasing absorption in the UV absorption spectrum indicates that 13 synthesized 2D AgBi perovskites hold indirect bandgaps, thus the optical bandgap is determined by fitting the variant Tauc equation

(Supplementary Figs. 6 and 7). The bandgaps of synthesized 2D AgBi perovskites are in the range of 1.84–1.99 eV, suggesting that the inorganic framework plays a dominant role in bandgap values of 2D perovskites and modifying organic spacers can further subtly modulate the electronic properties of 2D perovskites. In addition, a reported 2D RP phase perovskite with formula $(C_{10}S_2N_2H_{18})_2AgBiI_8$ is also collected as successful synthesis data[34].

## Subgroup discovery

Although datasets from high-throughput experiments contain both positive and negative material data, subjective preferences still exist due to idiosyncratic human choice and hard-to-control variables such as commercial availability. The subjective preferences reflect not only on the distribution of material synthesis data but also on the data that we can obtain. This can result in ML models that optimize and minimize global model errors based on prediction accuracy not being able to draw reliable conclusions, or ML models that perform well in specific subdomains but poorly on the entire dataset. In order to improve predictive accuracy and dig out reliable physicochemical insights, the biased distribution issue of the training set needs to be addressed. A promising solution is applying data-mining approaches to identify the applicable subdomains for ML models, then training ML models on the identified subdomain, demonstrating improved performance and more distinctive descriptors than models training on the whole biased dataset[35]. In practice, various ML techniques can be utilized to recognize subgroups of datasets, such as clustering and subgroup discovery[35,36]. Notably, the data distribution in the specific subdomain should be statically "most interesting", i.e., as large as possible while the target variable has the most distinctive distribution. Therefore, subgroup discovery is applied in this work to determine suitable subdomains for ML models to achieve the synthesis feasibility of 2D AgBi perovskites. Given a dataset for a specific challenge, the subgroup discovery approach can identify the subgroup with the most "informative distribution" and describe the identified subgroup in the form of "($f_1 < a$) and ($f_2 > b$) and …", where $f_i$ represents the $i$th descriptor, $a$ and $b$ represent the calculated threshold of corresponding descriptors, respectively[37]. As a descriptive technique, results obtained by subgroup discovery can be directly understood by human experts.

To develop high-performance ML models based on the subgroup discovery, appropriate material descriptors with respect to the target property are essential. Material synthesis is a complex process that depends not only on the kinetics and thermodynamic stability of materials itself, but also on the synthesis routes and the experimental

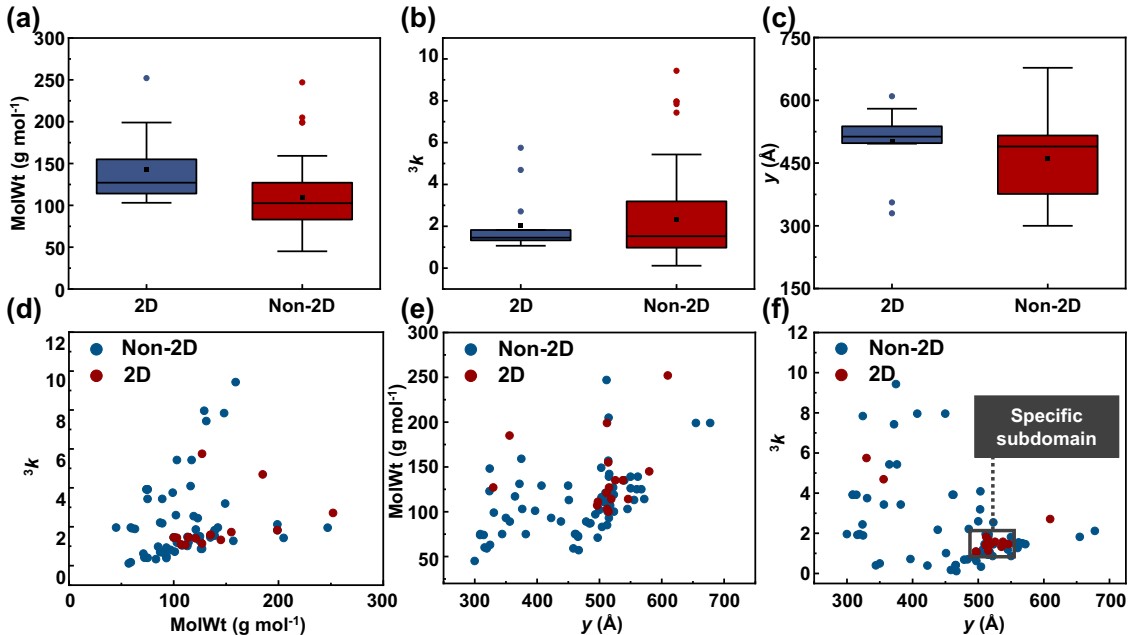

**Fig. 3 | Visualizing synthesis feasibility of 80 compounds with material descriptors.** Boxplots of synthesis feasibility of 80 compounds with (**a**) molecular weight (MolWt), (**b**) the third-order kappa shape index $^3k$, and (**c**) width $y$. In each boxplot, the central thick black line represents the median, color-shaded boxes represent the first and third quartiles (the 25th and 75th percentiles), and the whiskers extend no further than 1.5 times the distance between the first and third quartiles. **d**–**f** 2D projections of 3D scatter plot synthesis feasibility of 80 compounds with three important descriptors. In particular, the suitable range for $y$ and $^3k$ is marked by a black box. 2D and non-2D represent 2D perovskites and non-2D perovskites, respectively.

conditions such as synthetic methods, experimental parameters, and precursor species[30]. Note that the same synthesis method and parameters are utilized for high-throughput experiments in this work (Synthesis methods in Supplemental Methods, Supplementary Fig. 8), and the inorganic framework of all explored 2D HOIPs is AgBiI₈. Therefore, organic species featuring subtle structural and physicochemical characteristics, such as topological shape and size of molecules, are the most important variables to the synthesis feasibility of 2D HOIPs for a given inorganic framework. A set of common physicochemical descriptors obtained from the open-source cheminformatics package RDKit is first utilized to explore the quantitative structure-activity relationship (Supplementary Table 6)[38]. The distribution of features in the dataset is visualized as boxplots (Supplementary Figs. 9–12), where 50% of materials are located within the box (the lower and upper edges of the box represent the first and third quartile, respectively). In addition, the horizontal line in the box is the middle value of the dataset, and outliers distributed significantly differently from other data in the dataset are plotted as individual points outside the box. The data distribution results reveal that two descriptors stand out with a high correlation with the synthesis feasibility of 2D AgBi iodide perovskites, i.e., the molecular weight MolWt and the third-ordered kappa index $^3k$.

Moreover, the derivation of the rigid sphere model in our recent work has revealed that the width $y$ of organic spacers is critical for the structural stability of 2D HOIPs, consistent with the different distribution between $y$ of organic spacers in 2D perovskites and non-2D perovskites (Supplementary Fig. 12)[39,40]. 2D projections of this 3D data distribution map are generated, making scatter plots with reduced dimensions more suitable for human visualization ability. Red and blue plots in 2D projections correspond to organic spacers of 2D perovskites and non-2D perovskites (Fig. 3), respectively. Among these three projections, the distribution in ($y$, $^3k$) plane of organic spacers of non-2D perovskites is significantly different from that of 2D perovskites, in detail, molecules in the black box subdomain exhibit the most interesting distribution. The boundary of the subdomain is derived by utilizing the weighted relative accuracy (WRAcc), a popular

interestingness measurement in the subgroup discovery algorithm (Supplemental Methods). The WRAcc of subgroups with $y$ ranging from 486 to 550 pm and $^3k$ ranging from 1.01 to 1.89 is calculated (Supplementary Fig. 13), while $y$ and $^3k$ of the most interesting subdomain ranges from 496 to 546 pm and from 1.07 to 1.82, respectively. Notably, subtle change might occur among optimized molecular structures obtained by different basis sets[41], and this adds a tolerance region for the boundary of $y$.

Due to the constraint of molecular size, all molecules in the determined specific subdomain are based on the 5-membered or 6-membered ring, implying that cyclic organic spacers are more likely to stabilize the 2D AgBi perovskite structure than linear organic spacers. Recently, Wu et al. proposed that organic spacers with fewer branches and cycles are conducive to forming the 2D Pb perovskites[42]. The difference in preferred organic spacers between AgBi perovskites and Pb perovskites can be attributed to the inorganic framework. Our first-principle calculations reveal that the average metal-iodine bond length and metal-metal distance of PbI₄ are larger than those of AgBiI₈, indicating that the inorganic framework of AgBiI₈ consists of smaller octahedra, providing smaller semicuboctahedral cage for organic spacers (Supplementary Note 1, Supplementary Fig. 14, Supplementary Table 7). Moreover, the calculated Young's modulus of $(CH_3NH_3)_2AgBiI_6$ is higher than $CH_3NH_3PbI_3$, reflecting that the inorganic framework of AgBi perovskites exhibits lower softness. On the basis of the simplified model of perovskite lattice softness developed by Yin et al.[43], the enhanced modulus of AgBi perovskites originates from the reduced metal-halogen bond length. Therefore, the semicuboctahedral cage provided for organic spacers of 2D AgBi perovskites is not only small but also rigid. Linear organic spacers show high flexibility and diversity molecular conformations, which might damage the rigid inorganic framework of 2D AgBi perovskites, thereby further destabilize the 2D perovskite structure.

### Problem-specific descriptors
The distribution of 2D perovskites and non-2D perovskites is balanced in the determined specific subdomain, which contains 10 2D

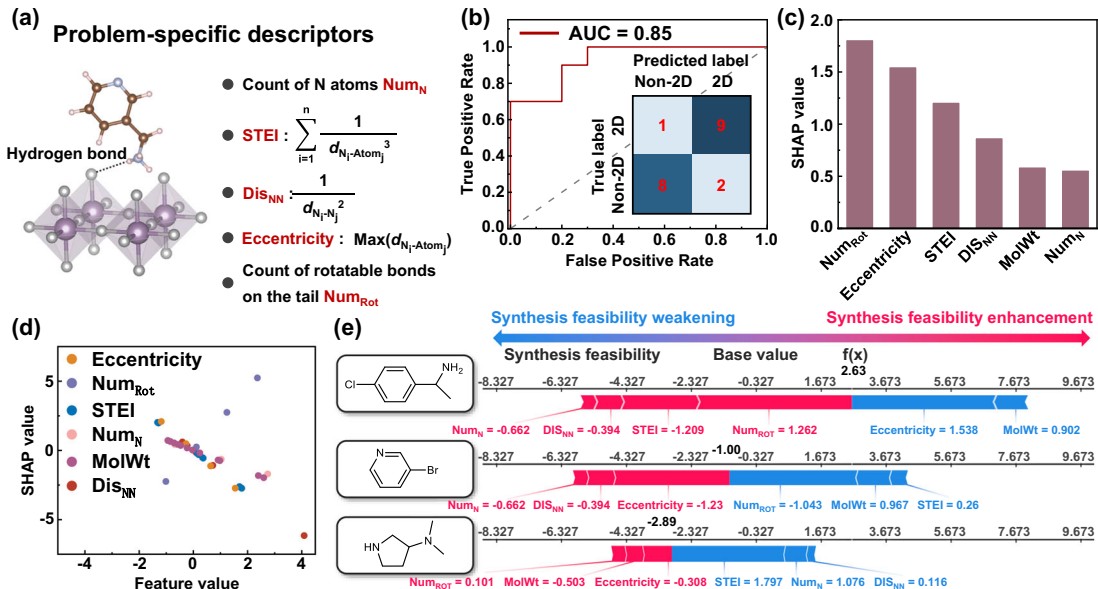

**Fig. 4 | Results and insights from ML model. a** Schematic sketch of the problem-specific descriptors. Here, $d_{N_i-Atom_j}$ represents the topological distance between the nitrogen i and atom j in the molecular skeleton, $d_{N_i-N_j}$ represents the topological distance between nitrogen i and j. **b** Receiver operating characteristic (ROC) curve and confusion matrix for the synthesis feasibility of 2D perovskites. **c** The sorted mean SHapley Additive exPlanations (SHAP) values of selected features in the ML model. **d** SHAP values for six features of the ML models, plots with different colors represent different features. **e** SHAP analysis of $(ClC_6H_4CH_4NH_3)_4AgBiI_8$, $(BrC_5H_4NH)_4AgBiI_8$, and $(NH_2C_4H_7NHC_2H_6)_2AgBiI_8$. Red and blue arrows represent the positive and negative contribution of features on the synthesis feasibility of 2D AgBi iodide perovskites, respectively. The expected base value of synthesis feasibility is $-0.0138$, and the ML-predicted synthesis feasibility of each sample is bolded in black.

perovskites and 10 non-2D perovskites (Supplementary Fig. 15). Note that three above features are insufficient for distinguishing 2D perovskites and non-2D perovskites in the specific domain, more distinctive descriptors related to the synthesis feasibility of 2D AgBi perovskites should be developed. The development of problem-specific descriptors is actually integrating physicochemical insights related to the specific problem at hand into ML models. To satisfy the requirements of high accuracy and convenience for prediction, material descriptors should bypass time-consuming first-principles calculations and be workable for target properties[10]. Therefore, although the dipole of organic spacers is highly correlated to the synthesis feasibility of 2D AgBi perovskites (Supplementary Fig. 12), four quantum chemical descriptors obtained from first-principles calculations are unadopted for training ML models. Accordingly, problem-specific descriptors are developed by utilizing the molecular graph theory, which is a useful tool for translating molecular structures into numerical topological indexes[44–46]. By disregarding hydrogen atoms to emphasize the molecular framework, the molecular topological structure can be extracted as a graph consisting of vertices and edges, where the vertices and edges represent atoms and chemical bonds, respectively.

Since 2D perovskites consist of alternately aligned organic and inorganic layers, the interaction between organic and inorganic components is a critical factor in the formation of 2D perovskite structure. The organic and inorganic components of 2D perovskites are linked by hydrogen bonds between amine groups of organic spacers and terminal halide of inorganic framework (Fig. 4a). Due to the different stacking modes between RP perovskites and DJ perovskites, RP perovskites also contain weak van der Waals interaction between adjacent organic layers. The stacking mode of 2D perovskites is attributed to the valence of organic spacers, which can be obtained by counting the number of nitrogen atoms $Num_N$. Moreover, the strength of hydrogen bonds is affected by the distance between bonding atoms and the local environment of bonding atoms, thus the distance between two nitrogen atoms $Dis_{NN}$, steric effect index (STEI) of nitrogen, and the number

of rotational bonds in the alkyl tail $Num_{Rot}$ are considered as problem-specific descriptors (Supplemental Methods, Supplementary Figs. 16 and 17). Note that the degree of molecular branching of organic spacers can influence the formability of 2D Pb perovskites[42,46], which can be described by the Eccentricity of organic spacer to some extent.

## ML classification model

Simple ML algorithms like support vector machine, linear regression, and gradient boosting are appropriate for modeling with small dataset[42,47]. We compared the performance of several common ML classification models on the identified subgroup, including logistic regression classification (LRC) model, decision tree classification (DTC) model, gradient boosting classification (GBC) model, and support vector classification (SVC) model (Supplementary Fig. 18). SVC model stands out for its classification accuracy among four ML classification models. Furthermore, the SVC algorithm also has the advantages of inherent simplicity and computation efficiency. Therefore, the SVC algorithm with the linear kernel is applied to develop the equation for the synthesis feasibility of 2D AgBi perovskites[48], which exhibits high interpretability and great predictive accuracy on the small-scale dataset[49]. The SVC model is trained by using 10-fold cross-validation in order to obviate the overfitting problem of the relatively small dataset (Supplemental Methods). The accuracy and the error of SVC models are assessed by employing the receiver operating characteristic (ROC) curve and confusion matrix[50,51]. The area under the ROC curve (AUC) of the SVC model is as high as 85%, meanwhile, only 1 out of 10 molecules of 2D perovskites is misclassified by the ML model, indicating the good performance of our trained ML model (Fig. 4b). On the basis of coefficients obtained from the training process of SVC model, the target property can be predicted as a sum of weighted feature inputs (Supplemental Methods). However, this equation is only suitable for the specific subdomain. To extend the applicable scope of this equation to the whole material space, the subgroup discovery and SVC model are combined to obtain the final equation for evaluating the

synthesis feasibility of 2D AgBi perovskites, as formulated

$$P = -1.98 \times \mathrm{Dis_{NN}} - 2.24 \times \mathrm{STEI} - 1.04 \times \mathrm{Eccentricity} - 1.58 \times \mathrm{Num_N}$$
$$+ 2.16 \times \mathrm{Num_{Rot}} - 0.03 \times \mathrm{MolWt} - \tan\left(\frac{\pi}{2} \times u\left(\left|\frac{y - 251}{25}\right| - 1\right)\right)$$
$$- \tan\left(\frac{\pi}{2} \times u\left(\left|\frac{3k - 1.445}{0.375}\right| - 1\right)\right) + 14.01, \quad u(x) = \begin{cases} 1, x > 0 \\ 0, x \leq 0 \end{cases}$$

(1)

Here, the value of $P$ indicates the synthesis feasibility of 2D AgBi iodide perovskites, which is easy to calculate. To test the robustness of the proposed equation, one sample in the training set is taken out and the remaining part of the dataset is utilized to train the SVC model. The procedure is repeated such that each sample in the training set is taken out once. Feature coefficients of most equations obtained from the trained SVC model are similar to the coefficients of the proposed equation, verifying the robustness and generalizability of the proposed equation (Supplementary Table 9). The combination of trigonometric function and step function is utilized to remove organic spacers not in this region. As the $P$ value increases, the synthesis feasibility of 2D AgBi perovskites increases, where 2D perovskite structure is expected to form for a determined range of $P > 0$. Moreover, the normalized coefficients of features are calculated for normalized features and listed in Table 1. Since the SVC model utilized in this work is a simplistic linear model, the contribution of features to the synthesis feasibility of compounds can be obtained by straightforward analyzed normalized coefficients. Positive feature coefficients indicate the

positive relationship between feature values and synthesis feasibility, and vice versa. Besides, the absolute values of normalized feature coefficients imply the importance of features, which are comparable to each other.

Utilizing model-agnostic interpretation strategies to extract meaningful physical and chemical insights from trained ML models has been proven to better understand ML predictions[10]. SHAP analysis[52], a popular strategy to interpret ML prediction results, is utilized in this work to explore the marginal contribution of individual descriptors and predict the synthesis feasibility of each sample (Supplemental Methods). As shown in Fig. 4c, $\mathrm{Num_{Rot}}$ is the most important feature to the synthesis feasibility of 2D AgBi perovskites, and the following features are the Eccentricity and STEI. Note that features related to the molecular topology exhibit a high correlation with the synthesis feasibility of 2D AgBi iodide perovskites. It is worth pointing out that the mean SHAP values ranking of selected features are different from the normalized coefficient obtained from the SVC model since the SHAP value reveals the marginal contribution of $i$th feature's addition calculated by $[f(S \cup \{i\} - f(S)]$, where $S$ represents all possible sets of the feature set. Compared to the model-dependent interpretation strategies, the advantages of SHAP analysis include not only sorting the importance of features but also indicating the negative or positive impact of each feature on the target property. The dependence between feature values and SHAP values is displayed in Fig. 4d, where different colors represent different features. The positive SHAP value means that the feature will drive the compound in the direction of high synthesis feasibility, while a negative SHAP value will push the prediction toward low synthesis feasibility. Note that $\mathrm{Num_{Rot}}$ is proportionate to the SHAP value, implying the lack of the alkyl tail is harmful to the synthesis of 2D AgBi iodide perovskites. Whereas other features are all inversely proportionate to the SHAP value, implying the small feature values are beneficial for the synthesis of 2D AgBi iodide perovskites. Taking three organic spacers as examples the local impact of six features is analyzed. As shown in Fig. 4e in bold, the predicted synthesis feasibility of $(\mathrm{ClC_6H_4CH_4NH_3})_4\mathrm{AgBiI_8}$, $(\mathrm{BrC_5H_4NH})_4\mathrm{AgBiI_8}$, and $(\mathrm{NH_2C_4H_7NHC_2H_6})_2\mathrm{AgBiI_8}$ is 2.42, −1.00, and −2.54, respectively, corresponding to one 2D perovskite and two non-2D perovskites, respectively. Features with red arrows are beneficial features to increase the synthesis feasibility of 2D AgBi iodide perovskites, and the length of arrows is proportional to SHAP values of given features. Conversely, features with blue arrows make negative contributions to 2D perovskite synthesis. Notably, $\mathrm{Num_{Rot}}$ makes the key negative contribution to the synthesis feasibility of $(\mathrm{BrC_5H_4NH})_4\mathrm{AgBiI_8}$, and the most negative feature for the synthesis feasibility of $(\mathrm{NH_2C_4H_7NHC_2H_6})_2\mathrm{AgBiI_8}$ is STEI. The lack of rotation bonds in the alkyl chain and the high steric hindrance effect of the nitrogen atom

**Table 1 | Feature coefficients of the equation for evaluating the synthesis feasibility of AgBi iodide perovskites**

|  | Unnormalized coefficient | Normalized coefficient |
|---|---|---|
| $\mathrm{Dis_{NN}}$[a] | −1.98 | −1.50 |
| STEI[b] | −2.24 | −1.53 |
| Eccentricity[c] | −1.04 | −1.76 |
| $\mathrm{Num_N}$[d] | −1.58 | −0.62 |
| $\mathrm{Num_{Rot}}$[e] | 2.16 | 2.21 |
| MolWt[f] | −0.03 | −0.76 |
| $C$[g] | 14.01 | −0.20 |

[a]$\mathrm{Dis_{NN}}$ represents the distance between two nitrogen atoms.
[b]STEI represents the steric effect index of nitrogen.
[c]Eccentricity represents the maximum distance between nitrogen and other atoms.
[d]$\mathrm{Num_N}$ represents the number of nitrogen atoms.
[e]$\mathrm{Num_{Rot}}$ represents the number of rotatable bonds on the tail of molecules.
[f]MolWt represents the molecular weight of molecules.
[g]$C$ represents the constant of of linear equation.

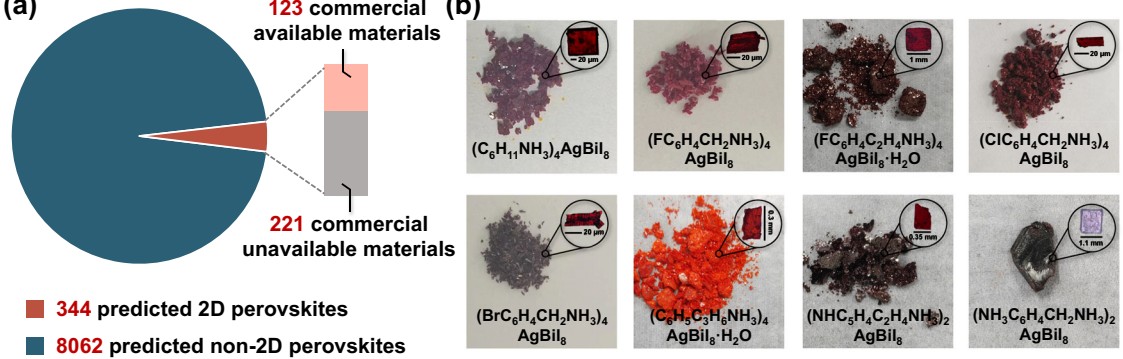

**Fig. 5 | Screening of 2D AgBi iodide perovskites with high synthesis feasibility and experiment validation. a** The schematic illustration of step-by-step screening for the prediction set. **b** The optical image of $(\mathrm{C_6H_{11}NH_3})_4\mathrm{AgBiI_8}$, $(\mathrm{FC_6H_4CH_2NH_3})_4\mathrm{AgBiI_8}$, $(\mathrm{ClC_6H_4CH_2NH_3})_4\mathrm{AgBiI_8}$, $(\mathrm{BrC_6H_4CH_2NH_3})_4\mathrm{AgBiI_8}$, $(\mathrm{FC_6H_4C_2H_4NH_3})_4\mathrm{AgBiI_8 \cdot H_2O}$, $(\mathrm{C_6H_5C_3H_6NH_3})_4\mathrm{AgBiI_8 \cdot H_2O}$, $(\mathrm{NHC_5H_4C_2H_4NH_3})_2\mathrm{AgBiI_8}$, and $(\mathrm{NH_3C_6H_4CH_2NH_3})_2\mathrm{AgBiI_8}$.

**Table 2 | Prediction and test results of 13 selected 2D perovskites**

| Compounds | ML-predicted results | Test results |
|---|---|---|
| $(C_6H_{11}NH_3)_4AgBiI_8$ | 2D perovskite | 2D perovskite |
| $(FC_6H_4CH_2NH_3)_4AgBiI_8$ | 2D perovskite | 2D perovskite |
| $(ClC_6H_4CH_2NH_3)_4AgBiI_8$ | 2D perovskite | 2D perovskite |
| $(BrC_6H_4CH_2NH_3)_4AgBiI_8$ | 2D perovskite | 2D perovskite |
| $(C_6H_5C_3H_6NH_3)_4AgBiI_8$ | 2D perovskite | 2D perovskite |
| $(FC_6H_4C_2H_4NH_3)_4AgBiI_8$ | 2D perovskite | 2D perovskite |
| $(NHC_5H_4C_2H_4NH_3)_2AgBiI_8$ | 2D perovskite | 2D perovskite |
| $(NH_3C_6H_4CH_2NH_3)_2AgBiI_8$ | 2D perovskite | 2D perovskite |
| $(CH_3C_6H_4CH_2NH_3)_4AgBiI_8$ | 2D perovskite | Non-2D perovskite |
| $(ClFC_6H_3CH_2NH_3)_4AgBiI_8$ | 2D perovskite | Non-2D perovskite |
| $(C_6H_5CH_2NH_3)_4AgBiI_8$ | 2D perovskite | Non-2D perovskite |
| $(C_6H_5C_2H_4NH_3)_4AgBiI_8$ | 2D perovskite | Non-2D perovskite |
| $(NH_3C_6H_4C_2H_4NH_3)_2AgBiI_8$ | 2D perovskite | Non-2D perovskite |

might weaken the strength of hydrogen bonding, resulting in the failure of 2D perovskite synthesis.

### Experiment validation

After training the ML model, the obtained equation is utilized to make a prediction for unexplored molecules. On the basis of molecular similarity related to molecules in the training and test sets, we collected 8406 molecules from the molecular database PubChem[53]. The high-dimensional representation of organic spacers is embedded into a 2D image by using the t-distributed stochastic neighbor embedding (t-SNE) method. For clarity, the ML-predicted synthesis feasibility and molecular structure of each point can be obtained by clicking the point in the 2D image (Supplementary Fig. 21, Supplementary Note 2, Supplementary Data 2). Successfully, 344 2D perovskites with high synthesis feasibility are screened out (Fig. 5a). However, since organic spacers in the prediction set were collected from the molecular database PubChem, commercial unavailability of some amines results in only 123 predicted 2D AgBi iodide perovskites hold the potential for further experimental synthesis (Supplementary Figs. 22–24). Since certain functional groups can react with HI[30], such as hydroxyl[54] and ether (Supplementary Fig. 25), nonreactive solvents or milder experimental conditions should be utilized when choosing organic spacers with these functional groups. To validate the reliability of our ML model, 13 commercially available organic spacers without hydroxyl and ether are unbiased selected and further examined via experiments (Table 2, Supplementary Fig. 26). As a result, 8 of 13 predicted 2D AgBi iodide perovskites with high synthesis feasibility are successfully synthesized, indicating that the success rate of ML-guided 2D AgBi iodide perovskites can reach 61.5%, which is much higher than the success rate based on the chemical intuition (16.4%). Note that synthesized single clear plank-shaped crystals are utilized to determine crystal structures, and the phase purity is verified by the powder X-ray diffraction (Supplementary Tables 10–13, Supplementary Fig. 27).

Moreover, the semiconducting properties of 8 selected 2D AgBi perovskites are further investigated by recording optical UV–vis spectra (Supplementary Fig. 28) and performing density functional theory calculations (Supplementary Fig. 29). These perovskites exhibit similar UV absorption curve and optical bandgaps relative to 2D AgBi perovskites in the training set, i.e., $(C_6H_{11}NH_3)_4AgBiI_8$ (1.93 eV), $(FC_6H_4CH_2NH_3)_4AgBiI_8$ (1.91 eV), $(ClFC_6H_3CH_2NH_3)_4AgBiI_8$ (1.89 eV), $(BrC_6H_4CH_2NH_3)_4AgBiI_8$ (1.87 eV), $(FC_6H_4C_2H_4NH_3)_4AgBiI_8·H_2O$ (1.76 eV), $(C_6H_5C_3H_6NH_3)_4AgBiI_8·H_2O$ (2.03 eV), $(NHC_5H_4C_2H_4NH_3)_2AgBiI_8$ (1.80 eV), and $(NH_3C_6H_4CH_2NH_3)_2AgBiI_8$ (1.93 eV). Their electronic structures show that the conduction band minimum (CBM) of 2D AgBi iodide perovskites is mainly dominated by

the hybrid of Bi $p$ orbital and I $p$ orbital, whereas the valence band maximum (VBM) is mainly from the Ag $d$ and I $p$ orbitals. The anisotropic interaction between Ag $d$ and I $p$ orbitals slightly incorporates Bi $s$ orbitals into the highest valence band, which enforces the location of VBM deviated from the $\Gamma$ point, leading to the indirect bandgap characteristic of 2D AgBi perovskites[55]. Moreover, analogous to traditional material $CH_3NH_3PbI_3$[56], organic molecules have no direct contribution to the band edge states of 2D AgBi perovskites. However, different organic spacers can influence the tilting and distortion of the inorganic framework via strong hydrogen bonding and further indirectly affect the electronic and optical properties of perovskites. Note that all synthesized 2D AgBi perovskites exhibit moderate bandgaps, which can serve as various optoelectronic devices. Furthermore, by appropriately modifying organic spacers of synthesized 2D AgBi perovskites in this work, more interesting characteristics such as anti-ferroelectrics can be modulated for the requirements of diversified functional materials.

## Discussion

In the above discussion, an approach that integrates high-throughput experiments, priori knowledge of chemistry, subgroup discovery, and SVC model is proposed to overcome the data sparsity and imbalance problem. Note that the data imbalance problem is common in many real-world problems, which has been considered one of the most important issues in training ML classification models. To date, many strategies have been proposed to address the data imbalance problem, such as under-sampling methods like CondensedNearestNeighbour and EasyEnsembleClassifier and over-sampling methods like synthetic minority oversampling technique (SMOTE)[10,21]. To comprehensively compare the performance of various methods, we unbiasedly selected ten compounds containing both 2D and non-2D perovskites in training and test sets for validation. As illustrated in Supplementary Table 14, three ML models (SMOTE, CondensedNearestNeighbour, and Easy-EnsembleClassifier) exhibit poor predictive ability on non-2D perovskites. In contrast, the ML model in this work is trained based on the identified specific subdomain, and validation results have demonstrated that our proposed integrated ML-based framework can well deal with this deficiency. More importantly, our proposed framework with some frozen experimental conditions can provide the probability estimates of synthesis feasibility of potential 2D HOIPs, which could also be further improved with optimization of experimental conditions, such as temperature, pressure, and solvent.

Note that our proposed framework is highly flexible and can integrate various other ML models with strong predictive power. For instance, alternative kernelized classification models with different kernel functions can be selected to distinguish 2D perovskites and non-2D perovskites in the specific domain. While many ML models have commendable predictive abilities, they often lack transparency in their predictions, making it difficult for humans to understand and extract physical and chemical insights. This lack of transparency hinders the development of new theories and insights. Therefore, it is essential to choose models that balance predictive accuracy with interpretability to facilitate the development of new theories and guide the discovery of advanced functional materials. Based on the Rashomon set argument[57], there is often existing at least one interpretable ML model with high predictive accuracy and interpretability. Knowledge gained from interpretable ML models can help to advance scientific understanding, which is fundamental to develop material science. Rather than creating models that are difficult to interpret such as SISSO, inherently interpretable ML models can provide more reliable explanations, which probably contain functions that can be approximated well by simpler functions related to priori knowledge. Besides, a set of informative features to quantify electronic, steric, and topological properties of organic precursors is proposed in this work (Supplemental Information), including common physicochemical

descriptors and problem-specific descriptors related to the specific problem at hand, which have great potential for use in developing ML models for subtle properties of HOIPs such as ferroelectric and chirality. Overall, by integrating appropriate ML techniques, physical and chemical insights, and high-throughput experiments, our proposed framework exhibits good extrapolating ability and interpretability, providing a promising avenue for future research in ML-aid synthesis of advanced functional materials and an in-depth understanding of 2D HOIP materials.

By integrating small-scale high-throughput experiments, physical and chemical insights, and ML techniques, we have developed an effective strategy to rapidly screen out 2D AgBi iodide perovskites with high synthesis feasibility. This strategy involves incorporating hydrogen bonding and subtle chemical interaction within 2D perovskite structures, alongside considering the typical physicochemical, steric, and topological properties of organic precursors. As part of our approach, we have defined a set of informative features that are closely associated with the synthesis feasibility of 2D AgBi perovskites. To solve the data imbalance problem, the subgroup discovery method is borrowed to discover the favorable formation region of 2D AgBi iodide perovskites. The trained ML model holds good performance with an accuracy of 85%, and the interpretable ML algorithm indicates that the molecular topology is critical for the synthesis of 2D AgBi iodide perovskites. Structure–property relationships reveal that cyclic organic spacers are more likely to stabilize the 2D perovskite structures than linear organic spacers. Low steric hindrance effect of nitrogen, fewer molecular branches, and rotational alkyl chains in cyclic organic spacers are beneficial for the synthesis of 2D AgBi iodide perovskites. Most importantly, an equation that can directly estimate the synthesis feasibility of 2D AgBi iodide perovskites is developed, and 344 molecules are identified as promising organic spacers of 2D AgBi perovskites from 8406 unexplored molecules under the guidance of this equation. Furthermore, to verify the predicted ability of our proposed equation, 13 predicted 2D perovskites are selected for experimental synthesis, and 8 compounds are successfully synthesized (61.5%). This study not only provides a practical way to rapid discovery of promising advanced functional materials but also a universal ML-aided synthesis framework that merges strong predictive capability with physicochemical interpretability.

## Methods

### Synthesis method and experimental characterization
Compounds in synthesis experiments were prepared by utilizing the evaporation method, the synthetic chemical reagents are reagent grade and are not further purified when used. The crystal structure of synthesized single clear crystals was determined by a single-crystal X-ray diffractometer, and the purity of bulk phases was confirmed by PXRD measurements. The semiconducting properties of synthesized 2D perovskites were investigated by measuring UV–vis diffuse reflectance spectroscopy, and optical bandgaps were determined by fitting the variant Tauc equation. The optical image of synthesized perovskites was acquired by employing a polarizing microscope. Details about synthesis methods and experimental characterization are given in the Supplementary Information.

### ML techniques and DFT calculations
The most suitable subdomain for ML models to achieve the synthesis feasibility of 2D AgBi perovskites is determined by the subgroup discovery approach[37]. The SVC model with the linear kernel is applied to obtain the final equation for evaluating the synthesis feasibility of 2D AgBi perovskites[48]. To obviate the overfitting problem of the relatively small dataset, the 10-fold cross-validation is utilized. The marginal contribution of individual descriptors is explored by performing SHAP analysis[52]. The first-principle calculations for 2D perovskites were performed by using the Vienna Ab initio Simulation Package 5.4

(VASP)[58]. To accurately compute the electronic structures, the Heyd–Scuseria–Ernzerhof (HSE06) hybrid functional[59,60] was applied. Details about ML techniques and DFT calculations are given in the Supplementary Information.

## Data availability
The data presented in this study are available in the manuscript file, the Supplementary Information files, and Source Data files. Source data are provided with this paper.

## Code availability
Data generated in this study and codes are available at https://github.com/wuyileiiiii/2D_perovskite_synthesizability[61].

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

## Acknowledgements

This work was supported by the National Key Research and Development Program of China (grant 2022YFA1503103, 2022YFB3807200,

2021YFA1200700), the Natural Science Foundation of China (grant 22173019, 22033002, 92056112, T2321002), the Basic Research Program of Jiangsu Province (BK20222007), the Fundamental Research Funds for the Central Universities (grant 2242022R40072). We thank the National Supercomputing Center of Tianjin and the Big Data Computing Center of Southeast University for providing the facility support on the calculations.

## Author contributions

M.-G.J. and J.W. conceived this work. Y.W. proposed ML-aided synthesis framework with guidance from M.-G.J. and J.W., C-F.W. performed small-scale high-throughput experiments with guidance from Y.Z., C-F.W. and Q.J. performed experimental validation of predicted 2D AgBi perovskites. Q.Z. and S.L. developed ML models. Y.W. and X.G. performed DFT calculations. Y.W., C-F.W., M-G.J., Y.Z. and J.W. analyzed the data and co-wrote the manuscript, with input from the other authors.

## Competing interests

The authors declare no competing interests.
