## [Peer Review File · Nature Communications]

Universal machine learning aided synthesis approach of two-dimensional perovskites in a typical laboratoryREVIEWER COMMENTS

Reviewer #1 (Remarks to the Author):

The manuscript by Wu et al. reports an interesting machine learning process to discover and analyze two-dimensional halide perovskites. In particular, the machine learning-predicted results are experimentally verified to clearly provide the final crystallographically resolved structures. This Exp+ML+DFT study is highly recommended for designing new functional materials. I suggest publication in this high-profile journal. However, the authors are recommended to address the following points.

1. I did not find the correct form of machine learning codes and datasets in the manuscript. Please provide a GitHub link (or other related website address) in the revised manuscript for the codes and upload all the datasets. A snapshot of sample train/test datasets in the manuscript is recommended.
2. In the introduction, please discuss more available machine learning methods to address the imbalanced data and small data size issue; there are several effective machine learning algorithms and methods in the literature. Please compare them with the present method.
3. A similar question: the authors employ SVM for model construction, which is a rather traditional machine learning algorithm. Please discuss the possibility of using other machine learning algorithms for prediction.
4. The molecular topology is critical for 2D perovskites. However, the topology is a broad concept and I suggest: (1) Be more accurate and specific in the conclusion (e.g., which topology is beneficial). (2) Provide a figure intuitively presenting the topology to help readers. Similarly, please summarize which chemical interactions and hydrogen bonds are beneficial for 2D perovskites revealed in the machine learning process.
5. Figure 4b: the confusion matrix demonstrates a limited number of data in the test set, which may make the machine learning model less convincing. Please comment on this.
6. typos and English:
 - (i) is explore by → is explored by
 - (ii) English: please correct the sentence "In this work, exemplified..."
 - (iii) Please move some of the contents in the first paragraph of "results or discussion" to the "background" or "method" section.

Reviewer #2 (Remarks to the Author):

Wu and coworkers show here a machine learning tool nurtured by experimental data, aimed at aiding to the synthesis of 2D perovskite-based materials, showing a clear improvement on the success ratio, going well beyond the chemical intuition. To my knowledge, this must be among the first experimental-based ML tools, and applied on perovskite materials, which are a hub of research materials with applications in different fields, and so, matching the journal scope. I am inclined towards recommendation, although authors should pay attention to the following points before a final decision is made.

1. Style and typos: In the abstract 'By application to...'. Used Oxford comma. Put foreign wording in italics, like i.e., via, etc.... It is 'van der Waals'. Put consistently orbitals in italics. Put Gamma in symbol, and in bold, being a vector. Use systematically EasyEnsemble for consistency. Put 'Conclusions'.
2. Would having 14 positive samples, and 66 negative ones, imply the ML would be biased towards the description of negative values?

3. It is unclear how the 79 amines were picked up by chemical intuition. Authors need to elaborate how, and based on what, the guesses were made. Couldn't these be guided by computational simulations?
4. When using SGD, authors need to better explain how that works.
5. Authors carried out high-throughput synthesis. The immediate question here is the reproducibility. To what extent is the synthesis reproducible? What about the possible impact of variations within the accuracy limits on the synthesis conditions?
6. In Figure 3, authors could well plot domain in a 3D view. What about clustering of 2D and non-2D perovskites on the $\gamma/3k/\text{MolWt}$ domain, as done, e.g. in <https://doi.org/10.1021/acscatal.2c01562>. Would that deliver the prototypical features for both sets?
7. When discussing rate of success of ML (66.7%) vs. chemical intuition (16.4%), it would be nice to highlight the percentage of a brute force, random pick up. It is worth highlighting as well that ML guidance multiplies by 4 the rate of success.

Reviewer #3 (Remarks to the Author):

In the manuscript "Universal machine learning aided synthesis approach in typical laboratory: a case study of two-dimensional perovskites" the authors present a framework of high throughput experiments and machine learning to guide the synthesis of a class of materials, in this case, silver/bismuth organic-inorganic hybrid perovskites.

Guided synthesis is of high significance. In the present paper, authors have tried to synthesize 80 perovskites, of which 14 were positive. Based on this database, a machine learning model was made and used to predict which organic spacers would result in a perovskite. In the end, 4 out of 6 predicted perovskites with high synthesis feasibility were successfully synthesized.

A similar approach in the literature exists in Ref. 37 of the manuscript. There the database was from the literature search (there are much more works on lead halide perovskites), but the idea and model are rather similar.

The main problem for the success of machine learning models is the number and quality of data. This study relies on only 80 data points and, therefore, it has to select only a few relevant parameters. The test set is much smaller. It would be better if the author tried some kind of active learning: after the initial model they could have started another high throughput synthesis with predicted perovskites, then retrained the model, and stop when the model gives accurate results. Statistically, it is not so unexpected that for the random selection of 6 organic spacers, there would be 4 successfully synthesized perovskites. At least the test set should be significantly larger to consider the results to be significant for the field and related fields.

I did not find methodological problems and it seems that there are enough details to reproduce the work.

In conclusion, I think that the presented framework could be useful in guiding the synthesis of 2D perovskites, but the present state does not seem to represent important advances compared to existing literature (see eg. Ref 37) given the small dataset both for training and testing.

Review #1:

The manuscript by Wu et al. reports an interesting machine learning process to discover and analyze two-dimensional halide perovskites. In particular, the machine learning-predicted results are experimentally verified to clearly provide the final crystallographically resolved structures. This Exp+ML+DFT study is highly recommended for designing new functional materials. I suggest publication in this high-profile journal. However, the authors are recommended to address the following points.

Reply:

We thank the reviewer for the positive remarks about our work and the helpful comments to further improve our manuscript. According to the reviewer's suggestions, we have made significant improvement in our manuscript. Below is our reply to the comments.

1. *I did not find the correct form of machine learning codes and datasets in the manuscript. Please provide a GitHub link (or other related website address) in the revised manuscript for the codes and upload all the datasets. A snapshot of sample train/test datasets in the manuscript is recommended.*

Reply:

Thank the reviewer for the suggestion. All codes and material data in this manuscript are now available at https://github.com/wuyilei1111/2D_perovskite_synthesizability.

We have also provided optical images, patterns of the powder X-ray diffraction, crystal data, and crystallographic data for 2D perovskites synthesized in this manuscript (Figure 5b, Figure S1-S7, S25-S26, Table S1-S5, S10-S13). Since outcomes of failed synthesis experiments are diverse and complex, such as 0D perovskites and amine iodide, further experimental characterizations are not performed.

2. *In the introduction, please discuss more available machine learning methods to address the imbalanced data and small data size issue; there are several effective machine learning algorithms and methods in the literature. Please compare them with the present method.*

Reply:

Class imbalance problem is common in many real-world problems, which has been considered as one of the most important issues in training ML classification models. To date, mainstream approaches to mitigate the class imbalance problem primarily contain over-sampling and under-sampling method. Over-sampling method achieves class balance by generating new samples in the minority class, while under-sampling method focuses on the reduction of samples in the majority class (Data Mining and Knowledge Discovery Handbook. Springer, 875-886). Various ML algorithms have been proposed to deal with class imbalance problem based on these two methods. For instance, synthetic minority oversampling technique (SMOTE) is developed based on the over-sampling method, whereas CondensedNearestNeighbour and EasyEnsembleClassifier are developed based on the under-sampling method (IEEE Transactions on Systems, Man, and Cybernetics, Part B (Cybernetics), 2009, 39(2), 539–550, <https://imbalanced-learn.org/>).

To comprehensively compare the performance of various methods, we unbiasedly selected ten compounds containing both 2D and non-2D perovskites in training and test sets for validation. As illustrated in Table R1, three ML models (SMOTE, CondensedNearestNeighbour, and EasyEnsembleClassifier) exhibit poor predictive ability on non-2D perovskites. Our strategy employed subgroup discovery techniques to identify the most informative subgroup around the

special data distribution in $(y, {}^3k)$ plane. Within the identified optimal subgroup, the count of samples in the negative and positive class is balanced, effectively mitigating the class imbalance problem in our dataset. Then the optimal subgroup is further applied to train ML models, which exhibits good predictive ability on both 2D perovskites and non-2D perovskites.

Table R1. Validation results of ML models based on various methods for imbalanced dataset. ML^{SMOTE} , ML^{Con} , and ML^{EEC} represents SMOTE, CondensedNearestNeighbour, and EasyEnsembleClassifier method, respectively.

	Test	ML^{SMOTE}	ML^{Con}	ML^{EEC}	$ML^{our\ work}$
$(C_5H_{11}N_3)_2AgBiI_8$	2D	Non-2D	Non-2D	2D	2D
$(C_6H_7NCl)_4AgBiI_8$	2D	2D	2D	2D	2D
$(C_4H_9NF_3)_4AgBiI_8$	2D	Non-2D	2D	2D	Non-2D
$(C_8H_{20}N_2)_2AgBiI_8$	Non-2D	2D	2D	2D	Non-2D
$(C_8H_{11}NBr)_4AgBiI_8-1$	Non-2D	2D	2D	2D	Non-2D
$(C_8H_{11}NBr)_4AgBiI_8-2$	Non-2D	2D	2D	2D	Non-2D
$(C_5H_8NS)_4AgBiI_8$	Non-2D	Non-2D	2D	2D	Non-2D
$(C_5H_5NI)_4AgBiI_8$	Non-2D	2D	2D	2D	2D
$(C_2H_7NBr)_4AgBiI_8$	Non-2D	Non-2D	2D	2D	Non-2D
$(C_5H_5NBr)_4AgBiI_8$	Non-2D	2D	2D	2D	2D

The inherent problems in small-scale dataset such as class imbalance and low feature dimensions might lead to the inferior performance of ML models. Common ML algorithms struggle to achieve satisfactory performance on our experimental synthesis dataset, thus we employed subgroup discovery to address the class imbalance problem. Then we compared performance of several common ML classifiers on the identified subgroup, including logistic regression classifier (LRC), decision tree classifier (DTC), gradient boosting classifier (GBC), and support vector classifier (SVC) (Figure R1). SVC stands out for its classification accuracy among four ML classifiers. Furthermore, SVC also has the advantages of inherent simplicity and computation efficiency. Therefore, SVC was utilized in this work.

Figure R1. Model performance of (a) logistic regression classifier, (b) decision tree classifier, (c) gradient boosting classifier, and (d) supporting vector classifier in our work.

Revisions made:

We added a discussion related to available ML techniques for imbalanced data in Page 4, paragraph 1 in the revised manuscript:

“Small datasets and imbalanced data distributions can easily bring about serious issues like overfitting, underfitting, and limited extrapolating abilities of ML models.^{19,20} Several strategies have been proposed to address class imbalance problem based on over-sampling and under-sampling method.²¹ Although there are numerous attempts to address these challenges, a comprehensive ML framework suitable for unfaithful dataset in material science has not yet been established.”

We added Table R1 as Table S14. We also revised discussion in Page 19, paragraph 2 in the revised manuscript:

“To date, many strategies have been proposed to address the data imbalance problem, such as under-sampling methods like CondensedNearestNeighbour and EasyEnsembleClassifier and over-sampling methods like synthetic minority oversampling technique (SMOTE).^{10,21} To comprehensively compare the performance of various methods, we unbiasedly selected ten compounds containing both 2D and non-2D perovskites in training and test sets for validation. As illustrated in Table S14, three ML models (SMOTE, CondensedNearestNeighbour, and EasyEnsembleClassifier) exhibit poor predictive ability on non-2D perovskites.”

We added Figure R1 as Figure S16. We also revised discussion in Page 13, paragraph 3 in the revised manuscript:

“Simple ML algorithms like supporting vector machine, linear regression, and gradient boosting are appropriate for modeling with small dataset.^{39,47} We compared performance of several common ML classification models on the identified subgroup, including logistic regression classification (LRC) model, decision tree classification (DTC) model, gradient boosting classification (GBC) model, and support vector classification (SVC) model (Figure S16). SVC

model stands out for its classification accuracy among four ML classification models. Furthermore, SVC algorithm also has the advantages of inherent simplicity and computation efficiency.”

3. *A similar question: the authors employ SVM for model construction, which is a rather traditional machine learning algorithm. Please discuss the possibility of using other machine learning algorithms for prediction.*

Reply:

Detailed comparison results among various ML models and related discussions are given in the reply to the above comment. We adopted four ML algorithms, namely, LRC, DTC, GBC, and SVC, to develop ML classification model on our experimental synthesis dataset (Figure R1). In particular, DTC and SVC exhibit good model performance, holding the potential to obtain reliable predictions. Due to the advantages of good performance and inherent simplicity (Adv. Mater. 2022, 34, 2202911, ACS Nano 2019, 13, 6308–6318, J. Am. Chem. Soc. 2017, 139, 17870–17881, Proc. Natl Acad. Sci. 2016, 113, 13588–13593), SVC was utilized in this work.

4. *The molecular topology is critical for 2D perovskites. However, the topology is a broad concept and I suggest: (1) Be more accurate and specific in the conclusion (e.g., which topology is beneficial). (2) Provide a figure intuitively presenting the topology to help readers. Similarly, please summarize which chemical interactions and hydrogen bonds are beneficial for 2D perovskites revealed in the machine learning process.*

Reply:

We thank the reviewer for his/her constructive suggestions. Structure-property relationships revealed by ML indicate that cyclic organic spacers are more likely to stabilize the 2D AgBi iodide perovskite structure than linear organic spacers. Low steric hinderance of nitrogen, less molecular branches, and rotational alkyl chains in cyclic organic spacers are beneficial for the synthesis of 2D AgBi iodide perovskites. The corresponding part of the conclusion has been revised.

To intuitively illustrate the definition of descriptors related to molecular topology in this work, we have taken 3-(aminomethyl)pyridine as an illustrative example (Figure R2). Hydrogens in molecules are ignored to emphasize the molecular skeleton. According to the molecular graph theory, molecular skeleton can be translated as a graph containing vertexes and edges, where vertexes and edges represent atoms and chemical bonds, respectively. This pattern of connectivity of constituent atoms in molecules is called molecular topology, has been used for mathematical characterization of molecular structures and establishment of structure-property relationship. Four problem-specific descriptors, namely, D_{NN} , Num_{Rot} , $Eccentricity$, and $STEI$, depending heavily on the distance matrix of molecules. Distance matrix is derived from the linkage of constituent atoms in the molecular skeleton, each matrix element d_{ij} in which represents the shortest topological distance between atom i and j in the molecular skeleton. The distance between two nitrogen atoms is defined as $D_{NN} = \frac{1}{d_{N_i-N_j}^2}$, where $d_{N_i-N_j}$ represents the topological distance between the nitrogen i and j . The number of rotational bonds in the alkyl tail Num_{Rot} is defined as the shortest topological distance between nitrogen and atoms on the ring. $Eccentricity$ is defined as the maximum topological distance between nitrogen and other atoms. The steric effect index of nitrogen is defined as $STEI = \sum_{i=1}^n \frac{1}{d_{N_i-Atom_j}^3}$, where $d_{N_i-Atom_j}$ is the topological distance between nitrogen i and the atom j in

the molecular skeleton.

Third-order kappa shape index 3k is calculated based on the graph path of molecules, which can be obtained by decomposing the molecular topological structure. Examples of one-, two-, and three-path fragments are given in the Figure R2. The definition of 3k is based on the count of non-hydrogen atoms and three-path fragments, in addition, size contribution of different atoms is considered. The ratio of covalent radii defined as $\alpha_i = (r_i/r_{\text{CSP}^3}) - 1$ is utilized to evaluate the difference of size contribution between atom i and $\text{C}(\text{sp}^3)$, and the alpha value α of molecules is the sum of α_i for each atom. Moreover, alpha values from covalent radii for common atoms with different valence states are provided in Table R2.

Figure R2. Schematic sketch of the molecular topology. Here, $d_{N_i-Atom_j}$ represents the topological distance between the nitrogen i and atom j in the molecular skeleton, $d_{N_i-N_j}$ represents the topological distance between nitrogen i and j , $d_{N_i-Atom_j}$ represents the topological distance between the nitrogen i and atom j in the ring.

Table R2. Alpha value from covalent radii.

Atom valence state	r (Å)	α
C (sp ³)	0.77	0.0
C (sp ²)	0.67	-0.13
C (sp)	0.60	-0.22
N (sp ³)	0.74	-0.04
N (sp ²)	0.62	-0.20
N (sp)	0.55	-0.29
O (sp ³)	0.74	-0.04
O (sp ²)	0.62	-0.20
F	0.72	-0.07
P (sp ³)	1.10	0.43
P (sp ²)	1.00	0.30
S (sp ³)	1.04	0.35

S (sp ²)	0.94	0.22
Cl	0.99	0.29
Br	1.14	0.48
I	1.33	0.73

Revisions made:

We revised the equation of third-order kappa shape index in Figure S14 and added Figure R2 as Figure S15. We added Table R2 as Table S8. We also revised discussion in problem-specific descriptors in Page 11, paragraph 2 in the Supporting Information:

“In addition, size contribution of different atoms is also considered and evaluated by the ratio of covalent radii between atom i and C(sp³), which is defined as $\alpha_i = (r_i/r_{CSP3}) - 1$ (Table S8).”

We revised the discussion related to molecular topology in Page 21, paragraph 2 in the revised manuscript:

“Structure-property relationships reveal that cyclic organic spacers are more likely to stabilize the 2D perovskite structures than linear organic spacers. Low steric hindrance effect of nitrogen, less molecular branches, and rotational alkyl chains in cyclic organic spacers are beneficial for the synthesis of 2D AgBi iodide perovskites.”

5. *Figure 4b: the confusion matrix demonstrates a limited number of data in the test set, which may make the machine learning model less convincing. Please comment on this.*

Reply:

We agree with the reviewer that dataset in this work contains a limited number of samples, which might lead to the concern of the trustworthiness of ML models. Therefore, we employed simple SVC algorithm to develop ML classification model. Moreover, cross-validation that is applicable to small dataset is utilized to evaluate the performance of SVC models. In addition, to test the robustness of proposed equation, one sample in training set is taken out and the remaining part of dataset is utilized to train SVC model. The procedure is repeated such that each sample in training set is taken out once. All models exhibit good performance, and feature coefficients of most equations obtained from trained SVC model are similar to coefficients of the proposed equation, verifying the robustness and generalizability of the proposed equation (Figure R3 and Table S9).

Figure R3. Model performance of trained SVC models on the dataset without (a) $\text{CC}(\text{C1}=\text{CC}=\text{C}(\text{C}=\text{C1})\text{Cl})\text{N}$, (b) $\text{CC}(\text{C1}=\text{CC}=\text{C}(\text{C}=\text{C1})\text{Br})\text{N}$, (c) $\text{CC1}=\text{CC}=\text{C}(\text{C}=\text{C1})\text{C}(\text{C})\text{N}$, (d) $\text{C1}=\text{CC}(\text{CC}=\text{C1N})\text{Cl}$, (e) $\text{C1}=\text{CC}(\text{CN}=\text{C1})\text{Br}$, (f) $\text{CC}(\text{C})\text{C1}=\text{CC}=\text{C}(\text{C}=\text{C1})\text{N}$, (g) $\text{CN}(\text{C})\text{C1}=\text{CC}=\text{CC}=\text{C1}$, (h) $\text{C1}=\text{CC}(\text{CN}=\text{C1})\text{I}$, (i) $\text{C1CC}(\text{CNC1})\text{F}$, (j) $\text{C1CC}(\text{CCC1N})\text{N}$, (k) $\text{C1CNCC1}(\text{F})\text{F}$, (l) $\text{C1CNCCC1}(\text{F})\text{F}$, (m) C1CNCCC1CN , (n) $\text{C1}=\text{C}(\text{NC}=\text{N1})\text{CCN}$, (o) $\text{CN}(\text{C})\text{C1CCNC1}$, (p) $\text{C1CCNC}(\text{C1})\text{N}$, (q) $\text{C1CN}(\text{CN1})\text{O}$, (r) $\text{C1CCN}(\text{CC1})\text{N}$, (s) C1CSCCN1 , and (t) $\text{C1CS}(\text{=O})(\text{=O})\text{CCN1}$.

6. *typos and English:*

(i) *is explore by* → *is explored by*

(ii) *English: please correct the sentence “In this work, exemplified....”*

(iii) *Pease move some of the contents in the first paragraph of “results or discussion” to the “background” or “method” section.*

Reply: We corrected these mistakes and carefully checked the whole manuscript.

Review #2:

Wu and coworkers show here a machine learning tool nurtured by experimental data, aimed at aiding to the synthesis of 2D perovskite-based materials, showing a clear improvement on the success ratio, going well beyond the chemical intuition. To my knowledge, this must be among the first experimental-based ML tools, and applied on perovskite materials, which are a hub of research materials with applications in different fields, and so, matching the journal scope. I am inclined towards recommendation, although authors should pay attention to the following points before a final decision is made.

Reply: We thank the reviewer for the positive remarks about our work and for the critical comments to further improve our manuscript. Below is our reply to the comments.

1. *Style and typos: In the abstract ‘By application to...’. Used Oxford comma. Put foreign wording in italics, like i.e., via, etc.... It is ‘van der Waals’. Put consistently orbitals in italics. Put Gamma in symbol, and in bold, being a vector. Use systematically EasyEnsemble for consistency. Put ‘Conclusions’.*

Reply: We carefully checked the whole manuscript and corrected those improper style and typos.

2. *Would having 14 positive samples, and 66 negative ones, imply the ML would be biased towards the description of negative values?*

Reply:

In this study, the class imbalance problem posed challenges for ML algorithms, as they tended to favor describing negative values within our dataset. To deal with this issue, we employed subgroup discovery techniques to determine the boundary of the most interesting distribution in (y, ³k) plane of organic spacers. The identified optimal subgroup achieves a balance between positive and negative classes, enabling the ML model to obtain a balanced representation for both classes. ML models trained on the identified subgroup exhibited good predictive ability and no bias towards the negative class. By utilizing subgroup discovery techniques, we effectively alleviated the bias caused by the class imbalance problem in this study.

3. *It is unclear how the 79 amines and were picked up by chemical intuition. Authors need to elaborate how, and based on what, the guesses were made. Couldn't these be guided by computational simulations?*

Reply:

Our synthesis experience and previously works related to 2D perovskites have revealed some chemical intuition on what kinds of amines might stabilize 2D perovskite structure (Chem. Rev. 121, 2230-2291 (2021)). (i) Most of 2D perovskites incorporate monovalent and divalent organic cations, since trivalent and tetravalent organic spacers are unlikely to achieve charge balance in 2D perovskites, which leads to the formation of low-dimensional structures. (ii) The shape of organic cations can be linear, branched, or cyclic (both aliphatic and aromatic), in which linear organic spacers are most conducive to form the 2D perovskites. (iii) The size of organic cations must fit in the maximum space given by inorganic framework. (iv) The alkyl chain of organic spacers should have suitable size and shape to fit in and form hydrogen bonds with the inorganic framework. Thus, the primary cations are the most favorable for 2D perovskites, followed by secondary, then tertiary, and quaternary ammonium cations ($\text{RNH}_3^+ > \text{R}_2\text{NH}_2^+ > \text{R}_3\text{NH}^+ > \text{R}_4\text{N}^+$). By considering amines utilized in previously reported 2D perovskites, above chemical intuition, and commercial availability of amines, 79 amines were selected.

Chemical intuition extracted from previously synthesis experience and works can guide the selection of amines by qualitatively evaluating the synthesis feasibility of 2D perovskites. Note that the synthesis feasibility of materials is difficult to quantitatively assess from the theoretical perspective. The thermal and thermodynamic stability of materials can be evaluated by performing ab initio molecular dynamic (AIMD) simulations and formation energy calculations, respectively. However, material synthesis is a complex process based not only on the thermodynamic stability but also kinetics, availability of precursors, and experimental conditions. Therefore, computational simulations usually exhibit a large gap with actual synthesis experiments. In this work, chemical intuition is transformed into features that can be quantitatively calculated and incorporated into ML

models to guide the synthesis experiments.

Revision made:

We revised the discussion in Page 7, paragraph 1 in the revised manuscript:

“Previous studies³⁰ and our extensive laboratory experience have provided valuable chemical intuitions into the selection of organic spacers that are conducive to forming the 2D perovskite structure. To satisfy the charge neutrality condition, monovalent and divalent organic spacers are generally incorporated into 2D perovskites. Furthermore, these organic spacers should have moderate size to fit in the inorganic framework of 2D perovskites. Linear and cyclic organic spacers, whether aliphatic or aromatic, are found to be favorable for the formation of 2D perovskite structure. Taking into account organic spacers employed in previously reported 2D perovskites, along with the chemical intuitions mentioned above, and commercial availability of amines, we have selected 79 promising amines for use in 2D AgBi iodide perovskite synthesis (Figure 2).”

4. *When using SGD, authors need to better explain how that works.*

Reply:

Subgroup discovery algorithm is a descriptive data mining technique that aims at identifying descriptions of the most informative subset of the whole dataset with respect to a target property. Descriptive language means that results obtained from ML models are directly interpretable by human experts. Material datasets for subgroup discovery can be divided into two parts, namely, description features A and target variables T. For instance, if we are interested in classifying metals and semiconductors, the target variable T is the bandgap and the descriptive features A are properties of materials such as ionic radii. Basic selectors can be developed as statements of descriptive features such as “the average ionic radii of samples are large” or “the sample exhibit rocksalt crystal structure”.

Figure R4. Schematic of (a) global ML classification model and (b) subgroup discovery. Gray line represents decision functions of global ML classification model, and gray cycle represents subdomain identified by subgroup discovery.

5. *Authors carried out high-throughput synthesis. The immediate question here is the reproducibility. To what extent is the synthesis reproducible? What about the possible impact of variations within the accuracy limits on the synthesis conditions?*

Reply:

To test experimental reproducibility, synthesis experiments for $(\text{NH}_2\text{C}_5\text{H}_8\text{F}_2)_4\text{AgBiI}_8$ were individually repeated ten times and 2D perovskites were successfully synthesized each time (Figure R5), suggesting the good reproducibility. Ten individual synthesis experiments of

$(\text{NH}_2\text{C}_5\text{H}_8\text{F}_2)_4\text{AgBiI}_8$ were utilized the same experimental conditions employed in the high-throughput synthesis experiments in the manuscript. The detailed experimental procedures and conditions can be found in the synthesis method section of the Supplemental Information. Thus, experimental outcomes can be feasibly replicated by applying experimental conditions provided in the manuscript.

Figure R5. Images of $(\text{NH}_2\text{C}_5\text{H}_8\text{F}_2)_4\text{AgBiI}_8$ synthesized in ten synthesis experiments.

We determined the optimal synthesis method and conditions for 2D AgBi iodide perovskites based on previous synthesis experience of our laboratory and preceding work. To minimize the impact of experimental conditions on synthesis outcomes, the same experimental conditions such as inorganic precursors, solvent, concentration, and temperature were utilized in practice. In addition, previous synthesis experience of our laboratory indicates that different experimental conditions might lead to different synthesis outcomes. For instance, only commercial hydroiodic acid with mass fraction of more than 55% can be used to synthesize double perovskites, and those with mass fraction of 47% leads to the formation of amine iodide or 0D perovskites. The synthesis of most double perovskites requires the condition that Ag^+ ions are more than Bi^{3+} ions to prevent the formation of yellow Bi^{3+} iodide phase. Therefore, adjusting experimental conditions such as temperature, pressure, and solvent might impact outcomes of synthesis experiments.

Revision made:

We added Figure R5 as Figure S3. We also revised the discussion in Page 2, paragraph 1 in the Supporting Information:

“To assess the experimental reproducibility of our synthesis experiments, we conducted ten individual repetitions of the synthesis process for $(\text{NH}_2\text{C}_5\text{H}_8\text{F}_2)_4\text{AgBiI}_8$. Remarkably, in each instance, we successfully synthesized 2D perovskites (Figure S3), indicating excellent reproducibility.”

6. In Figure 3, authors could well plot domain in a 3D view. What about clustering of 2D and non-2D perovskites on the $y/3k/\text{MolWt}$ domain, as done, e.g. in <https://doi.org/10.1021/acscatal.2c01562>. Would that deliver the prototypical features for both sets?

Reply:

Thank the reviewer for his/her constructive suggestion. We utilized k-means algorithm to cluster our dataset in two groups, namely, cluster1 and cluster2. Since k-means algorithm aims to divide samples into k disjoint clusters by minimizing within-cluster variances, the distance between

samples and cluster centroid is the critical factor for clustering. Clustering results reveal that most of 2D perovskites are classified into cluster2. Notably, cluster2 still encompasses a substantial presence of non-2D perovskites (Figure R6). Therefore, k-means algorithm encounters challenges in delivering prototypical features for both sets in our work.

Figure R6. 3D k-means clustering of synthesis feasibility of 80 compounds.

7. When discussing rate of success of ML (66.7%) vs. chemical intuition (16.4%), it would be nice to highlight the percentage of a brute force, random pick up. It is worth highlighting as well that ML guidance multiplies by 4 the rate of success.

Reply: Thank for the reviewer's helpful suggestion, we have highlighted the improvement in success rate of the synthesis feasibility under the guidance of ML in Abstract:

“Through application of our approach to challenging and consequential synthesis problem of 2D silver/bismuth organic-inorganic hybrid perovskites, we have increased the success rate of the synthesis feasibility by a factor of four relative to traditional approaches.”

Review #3:

In the manuscript "Universal machine learning aided synthesis approach in typical laboratory: a case study of two-dimensional perovskites" the authors present a framework of high throughput experiments and machine learning to guide the synthesis of a class of materials, in this case, silver/bismuth organic-inorganic hybrid perovskites.

Guided synthesis is of high significance. In the present paper, authors have tried to synthesize 80 perovskites, of which 14 were positive. Based on this database, a machine learning model was made and used to predict which organic spacers would result in a perovskite. In the end, 4 out of 6 predicted perovskites with high synthesis feasibility were successfully synthesized.

I did not find methodological problems and it seems that there are enough details to reproduce the work.

Reply: We thank the reviewer for the positive remarks about our work and for the critical comments to further improve our manuscript. Below is our reply to the concerned comments.

The main problem for the success of machine learning models is the number and quality of data. This study relies on only 80 data points and, therefore, it has to select only a few relevant parameters. The test set is much smaller. It would be better if the author tried some kind of active learning: after the initial model they could have started another high throughput synthesis with predicted perovskites, then retrained the model, and stop when the model gives accurate results.

Reply:

Due to the challenges of synthesis experiments, subjective preferences from idiosyncratic human choice, hard-to-control variables, commercial availability, and so on, these inevitably lead to the problem of insufficient and unbalanced data. Thus, it is essential to strike an optimal balance between datasets and algorithms. In our case, we applied subgroup discovery along with simple ML algorithm to effectively handle the class imbalance problem. The advantage of this approach lies in its ability to obtain accurate predictions without significantly increasing the cost of additional experiments. By integrating a *priori* knowledge of chemistry and subgroup discovery, our proposed ML framework shows good performance and significantly improves the success rate of experimental synthesis.

We agree with the reviewer that active learning can enhance model performance while minimizing the need for extensive labeled data. As discussed above, our proposed ML framework exhibits superior predictive ability, which remarkably improves about 45% success rate of the synthesis feasibility relative to traditional approaches. In addition, active learning is particularly suitable for the exploration of large chemical spaces, and the prediction set of our work is not particularly large, so general machine learning methods are sufficient. Therefore, the active learning strategy was not employed in this work. In future related research, we may carry out active learning to accelerate material synthesis.

Statistically, it is not so unexpected that for the random selection of 6 organic spacers, there would be 4 successfully synthesized perovskites. At least the test set should be significantly larger to consider the results to be significant for the field and related fields.

Reply:

We thank the reviewer for pointing this important issue. According to the suggestions of the reviewer, our experimental collaborators conducted additional experimental verifications. Now, we totally selected 13 ML-predicted 2D perovskites with commercially available organic spacers for experimental synthesis, and eight of them were successfully synthesized, resulting in a success rate of 61.5% (Figure R6, Table R3). Since the organic spacers in the prediction set were collected from the molecular database PubChem, some amines are not commercially available. Therefore, we are currently unable to proceed with further experimental validation, given the experimental resource and cost involved. We revised the discussion related to experimental validation results in the manuscript and Supporting Information.

Table R3. Prediction and test results of 13 selected 2D perovskites.

Compounds	ML-predicted results	Test results
$(\text{C}_6\text{H}_{11}\text{NH}_3)_4\text{AgBiI}_8$	2D perovskite	2D perovskite
$(\text{FC}_6\text{H}_4\text{CH}_2\text{NH}_3)_4\text{AgBiI}_8$	2D perovskite	2D perovskite
$(\text{ClC}_6\text{H}_4\text{CH}_2\text{NH}_3)_4\text{AgBiI}_8$	2D perovskite	2D perovskite

$(\text{BrC}_6\text{H}_4\text{CH}_2\text{NH}_3)_4\text{AgBiI}_8$	2D perovskite	2D perovskite
$(\text{C}_6\text{H}_5\text{C}_3\text{H}_6\text{NH}_3)_4\text{AgBiI}_8$	2D perovskite	2D perovskite
$(\text{FC}_6\text{H}_4\text{C}_2\text{H}_4\text{NH}_3)_4\text{AgBiI}_8$	2D perovskite	2D perovskite
$(\text{NHC}_5\text{H}_4\text{C}_2\text{H}_4\text{NH}_3)_2\text{AgBiI}_8$	2D perovskite	2D perovskite
$(\text{NH}_3\text{C}_6\text{H}_4\text{CH}_2\text{NH}_3)_2\text{AgBiI}_8$	2D perovskite	2D perovskite
$(\text{CH}_3\text{C}_6\text{H}_4\text{CH}_2\text{NH}_3)_4\text{AgBiI}_8$	2D perovskite	Non-2D perovskite
$(\text{ClFC}_6\text{H}_3\text{CH}_2\text{NH}_3)_4\text{AgBiI}_8$	2D perovskite	Non-2D perovskite
$(\text{C}_6\text{H}_5\text{CH}_2\text{NH}_3)_4\text{AgBiI}_8$	2D perovskite	Non-2D perovskite
$(\text{C}_6\text{H}_5\text{C}_2\text{H}_4\text{NH}_3)_4\text{AgBiI}_8$	2D perovskite	Non-2D perovskite
$(\text{NH}_3\text{C}_6\text{H}_4\text{C}_2\text{H}_4\text{NH}_3)_2\text{AgBiI}_8$	2D perovskite	Non-2D perovskite

Figure R7. Optical image, UV-vis absorption spectra, and patterns of the powder X-ray diffraction of (a) $(\text{C}_6\text{H}_5\text{C}_3\text{H}_6\text{NH}_3)_4\text{AgBiI}_8 \cdot \text{H}_2\text{O}$, (b) $(\text{FC}_6\text{H}_4\text{C}_2\text{H}_4\text{NH}_3)_4\text{AgBiI}_8 \cdot \text{H}_2\text{O}$, (c) $(\text{NHC}_5\text{H}_4\text{C}_2\text{H}_4\text{NH}_3)_2\text{AgBiI}_8$, and (d) $(\text{NH}_3\text{C}_6\text{H}_4\text{CH}_2\text{NH}_3)_2\text{AgBiI}_8$.

Revision made:

We revised the discussion in Page 17, paragraph 2 in the revised manuscript:

“Successfully, 344 2D perovskites with high synthesis feasibility are screened out (Figure 5a). However, since organic spacers in the prediction set were collected from the molecular database PubChem, commercial unavailability of some amines results in only 123 predicted 2D AgBi iodide

perovskites hold the potential for further experimental synthesis (Figure S20-S22). Since certain functional groups can react with HI,³⁰ such as hydroxyl⁵⁴ and ether (Figure S23), nonreactive solvents or milder experimental conditions should be utilized when choosing organic spacers with these functional groups. To validate the reliability of our ML model, 13 commercially available organic spacers without hydroxyl and ether are unbiased selected and further examined *via* experiments (Table 2, Figure S24).”

A similar approach in the literature exists in Ref. 37 of the manuscript. There the database was from the literature search (there are much more works on lead halide perovskites), but the idea and model are rather similar.

In conclusion, I think that the presented framework could be useful in guiding the synthesis of 2D perovskites, but the present state does not seem to represent important advances compared to existing literature (see eg. Ref 37) given the small dataset both for training and testing.

Reply:

We thank the reviewer for pointing this important issue, his/her constructive suggestion makes us think more deeply about the uniqueness of our work. **The purpose of our work is to provide a general and practical framework to accelerate material synthesis process**, more specifically, what we try to solve is how to directly assess the synthesis feasibility of unexplored materials based on small-scale experiments *via* machine learning methods. Synthesizable probability of 2D AgBi iodide double perovskites is investigated as a case study. Very differently, Ref. 37 used ML to classified the dimensionality of low-dimensional Pb iodide perovskites based on their **already synthesized perovskites**, that is, the prerequisite is that the materials should be synthesized first and the dimensionality can then be classified into “can form 2D” and “cannot form 2D”. Therefore, we do not think that these two works utilized similar ideas fundamentally. Below we will give a detailed discussion about the differences between our work and Ref. 37.

First of all, two works exhibit fundamental differences in terms of ideas. Work in ref. 37 aims to utilize ML techniques to classify the dimensionality of synthesized low-dimensional Pb iodide perovskites. Whereas our work aims to utilize ML techniques for the acceleration of material synthesis based on small dataset from typical laboratory, and the power of our proposed strategy is showcased on the experimental synthesis of 2D AgBi iodide perovskites. Thus, non-2D perovskites in our work contains more situations of failed synthesis experiments in practice.

Second, in terms of ML models, due to the inherent sparsity and imbalanced problem of experimental synthesis dataset, our works developed different ML framework to obtain high-performance ML models from Ref. 37. As discussed above, the limitation of experimental conditions leads to the imbalanced problem of our dataset, which highly impacts the performance of ML models. Note that data imbalanced problem is common in many real-world problems, which has been considered as one of the most important issues in training ML classification models. Our work dealt with imbalanced problem in dataset by utilizing **subgroup discovery techniques** in particular, and identified optimal subgroup with balanced data distribution is further applied to train ML models. Our strategy exhibits good performance and significantly improves the success rate of experimental synthesis.

Third, we have formulated a comprehensive set of material descriptors concerning the

synthesis feasibility of 2D AgBi iodide perovskites, drawing upon knowledge of physicochemical or mechanism *a priori* (Table S6). In fact, 2D AgBi iodide perovskite structures are very complicated, and some amines that meet the characteristic conditions for the formation of lead-based perovskites are not necessarily suitable for the formation of double perovskites (Matter 2019, 1, 465-480, J. Am. Chem. Soc. 2019, 141, 12880–12890). Chemical intuition extracted from experimental synthesis results are transformed into material descriptors which can be quantitatively calculated and incorporated into ML models. Overall, six of eight descriptors included in proposed equation of our work are derived from our physicochemical knowledge and previous research, especially two descriptors for subgroup discovery (Table R4).

Finally, we would like to make a small point, that is, structure-property relationship uncovered by ML in two works exhibits significant differences. For 2D Pb perovskites, organic spacers with less cycles are conducive to form the 2D perovskite structure. In contrast, cyclic organic spacers are more likely to stabilize the 2D AgBi iodide perovskite structures than linear organic spacers, which are instructive for the synthesis of 2D AgBi iodide perovskites. Furthermore, 21 new 2D AgBi iodide perovskites in total have been successfully synthesized in this work, which highly enrich the family of 2D AgBi perovskites.

In summary, the fundamental difference between our work and Ref. 37 is that used ML to determine the dimensions of perovskites, and our work is to determine the possibility that 2D perovskites can be synthesized. More importantly, our work provides a practical framework for acceleration problems of multidimensional chemical synthesis from typical laboratory with limited experimental resources available. Notably, subgroup discovery has been integrated into the ML-aided synthesis framework for the first time, which effectively addresses the class imbalance problem of experimental synthesis dataset.

Table R4. Key molecular descriptors in ref. 37 and this work.

	Descriptors
Ref. 37	STEI , eccentricity, large ring size, H donor count
Our work	3k , y , Dis_{NN} , STEI , eccentricity, Num_N , Num_{Rot} , MolWt

REVIEWERS' COMMENTS

Reviewer #1 (Remarks to the Author):

The authors answered my questions. "Accept" is my recommendation.

Reviewer #2 (Remarks to the Author):

Authors duly treated the raised points. The manuscript is publishable in its present form.

Reviewer #3 (Remarks to the Author):

The authors have addressed all my concerns in the revision. They have supported their conclusions with additional data. I recommend the manuscript for publication.